# Pharmacological Evaluation of Araliadiol as a Novel Anti-Inflammatory Agent in LPS-Induced RAW 264.7 Cells

**DOI:** 10.3390/biomedicines13061408

**Published:** 2025-06-08

**Authors:** Seokmuk Park, Suhyeon Cho, Hee-Jae Shin, Seyeol Baek, Hye-In Gwon, Jungmin Lee, Dae Sung Yoo, Han Woong Park, Dae Bang Seo, Seunghee Bae

**Affiliations:** 1Department of Biological Engineering, Konkuk University, 120 Neungdong-ro, Gwangjin-gu, Seoul 05029, Republic of Korea; ted968@konkuk.ac.kr (S.P.); woyg@konkuk.ac.kr (S.C.); shj02589@naver.com (H.-J.S.); ted3842@naver.com (S.B.); eoeh7599@naver.com (H.-I.G.); 2Department of Bio-Cosmetics Engineering, Sungkyul University, 53 Seonggyeoldaehak-ro, Manan-gu, Anyang-si 14097, Republic of Korea; iris-521@hanmail.net; 3ASK Company Co., Ltd., 86 Dongdaegu-ro, Suseong-gu, Daegu 706841, Republic of Korea; dsyoo@riman.com (D.S.Y.); hwpark@riman.com (H.W.P.); dbseo@riman.com (D.B.S.)

**Keywords:** anti-inflammation, araliadiol, cytokine, lipopolysaccharide, phytochemical

## Abstract

**Background/Objectives**: Inflammatory disorders contribute to the pathogenesis of numerous diseases and are known to markedly reduce quality of life. Although anti-inflammatory drugs approved by the Food and Drug Administration are available, their prolonged use is frequently associated with adverse effects. In this study, we evaluated the pharmacological properties of araliadiol, a naturally occurring polyacetylene compound, as a novel anti-inflammatory agent. **Methods**: An in vitro hyperinflammatory model was established by stimulating RAW 264.7 cells with lipopolysaccharide (LPS). Dexamethasone (DEX) was used as a positive control to compare anti-inflammatory efficacy. The protective effects of araliadiol against LPS-induced cytotoxicity were assessed using adenosine triphosphate content and crystal violet staining assays. The anti-inflammatory activity was further examined by quantitative reverse transcriptase–polymerase chain reaction, Western blotting, cell fractionation, immunofluorescence staining, a nitric oxide assay, and an enzyme-linked immunosorbent assay. **Results**: Araliadiol significantly attenuated cytotoxicity and cell death in LPS-stimulated RAW 264.7 cells. It suppressed the expression of cell death markers Cleaved caspase-3 and Cleaved PARP-1. In addition, araliadiol downregulated key pro-inflammatory mediators, including inflammasome-related genes, cytokines, chemokines, and inducible nitric oxide synthase. It also reduced the expression of Cox-2 and PGE_2_, indicating potential anti-hyperalgesic effects. Moreover, araliadiol inhibited the activation of Nfκb and Stat1 signaling pathways in LPS-stimulated macrophages. **Conclusions**: Araliadiol demonstrated robust anti-cytotoxic, anti-inflammatory, and anti-hyperalgesic activities in LPS-induced RAW 264.7 cells, with efficacy comparable to DEX. These findings support its potential as a plant-derived therapeutic candidate for the management of inflammatory conditions.

## 1. Introduction

Inflammatory responses are innate defense mechanisms triggered by internal or external harmful stimuli, such as pathogens and chemical irritants. While inflammation plays an essential role in maintaining physiological homeostasis, excessive or abnormal inflammatory responses can result in tissue damage and dysfunction [1]. Once the initiating stimuli are eliminated, the resolution of inflammation is essential for restoring tissue homeostasis. Failure to adequately resolve inflammation can result in chronic inflammation, which underlies diseases such as osteoarthritis, colitis, asthma, and psoriasis. These diseases also increase the risk of complications such as cancer and cardiovascular disease due to sustained tissue damage [2].

First-line therapies for inflammatory diseases include corticosteroids, non-steroidal anti-inflammatory drugs (NSAIDs), and monoclonal antibodies (mAbs). Corticosteroids are widely used because of their potent anti-inflammatory effects. They act by inhibiting pro-inflammatory gene expression via cis-repression, suppressing NFκB-mediated transcription through trans-repression, and enhancing the anti-inflammatory gene expression of Annexin-1 and IκB-α (trans-activation) [3]. Common synthetic corticosteroids are widely prescribed for conditions such as asthma, psoriasis, Addison’s disease, rheumatoid arthritis (RA), and inflammatory bowel disease (IBD) [4]. However, despite their effectiveness, the long-term use of corticosteroids is associated with a range of adverse effects [5]. Specifically, long-term use (≥60 days) has been linked to weight gain (70% of patients), cataracts (15%), fractures (12%), skin thinning, and sleep disturbances [6]. NSAIDs such as aspirin and ibuprofen inhibit cyclooxygenase (COX) enzymes, thereby reducing prostaglandin-mediated inflammation and pain. However, long-term NSAID use is associated with gastrointestinal side effects, such as nausea, dyspepsia, abdominal pain, and cramping, in 20–40% of users [7,8]. NSAIDs also pose risks of acute kidney injury, hypertension, and cardiovascular complications [8].

Recent advances in small-molecule and monoclonal antibody therapies targeting inflammatory biomarkers have broadened therapeutic options [9,10]. For instance, apremilast, a dual inhibitor of phosphodiesterase-4 and tumor necrosis factor (TNF)-α, has been introduced for the treatment of psoriasis and psoriatic arthritis [11], while infliximab, a chimeric monoclonal antibody, is utilized in autoimmune diseases such as Crohn’s disease, RA, and psoriasis [12]. Despite their therapeutic potential, monoclonal antibodies may cause adverse effects such as anaphylaxis, serum sickness, antibody production, and long-term safety concerns, such as increased risks of cancer and cardiotoxicity [13]. These limitations have led to increasing interest in plant-derived anti-inflammatory compounds with improved safety.

Phytochemicals—secondary metabolites from plants—have long served as sources of medicinal agents. Notably, nearly half of all drugs approved since 1994 have been derived from natural products, underscoring the significance of phytochemicals as valuable lead compounds for drug discovery [14]. Despite ongoing challenges related to isolation, purification, standardization, quality control, sustainable sourcing, intellectual property issues, and limited selectivity, phytochemicals offer several advantages, such as lower development costs, relatively high bioavailability, a favorable safety profile, and fewer side effects, that make them promising candidates for novel drug development [14,15,16,17,18,19]. Given the sustained interest in natural product-based therapeutics, this study investigated the pharmacological potential of araliadiol—a polyacetylene compound—as a candidate anti-inflammatory agent.

Araliadiol is abundantly present in *Centella asiatica* (*C. asiatica*) and *Aralia cordata*, both of which have been used extensively in traditional medicine for their anti-inflammatory properties [20,21,22]. Pentacyclic triterpenes, such as asiaticoside, madecassoside, asiatic acid, and madecassic acid, are widely recognized as the main bioactive compounds in *C. asiatica* [23]. However, despite extensive research on its anti-inflammatory properties, few studies have assessed whether these effects are exclusively due to triterpenes, and clinical evidence remains limited [23,24]. Given that *C. asiatica* extracts are commonly used in commercial products (e.g., Madecassol^®^) and that over 130 secondary metabolites have been identified, there is a strong rationale for exploring the anti-inflammatory potential of novel phytochemicals beyond the triterpene class [24,25]. Recently, increasing attention has been given to the biological activities of araliadiol (C_15_H_20_O_2_), a polyacetylene compound. Initially noted for its anticancer effects, it has also shown potential in metabolic disorders via enhanced glucose uptake, as well as antioxidant, neuroprotective, and anti-hair loss potential [26,27,28,29]. In particular, its antioxidant and cytoprotective effects under oxidative stress suggest promising anti-inflammatory potential [27]. As a natural constituent of *C. asiatica*, araliadiol warrants further investigation for its role in inflammation. This study, therefore, aims to assess its anti-inflammatory effects in LPS-stimulated RAW 264.7 macrophages and elucidate its underlying molecular mechanisms.

## 2. Materials and Methods

### 2.1. Chemicals and Cell Culture for RAW 264.7 Cells

Araliadiol was provided by ASK Company Co., Ltd. (Daegu, Republic of Korea). The isolation of araliadiol from *C. asiatica* extract was performed following the method described in a previous study [29]. Briefly, one kilogram of dried *C. asiatica* was extracted twice with 20 L of 70% methanol at 70 °C for 5 h. The combined supernatants were concentrated, filtered, and subjected to methanol removal, followed by partitioning between hexane and water. The hexane layer was concentrated and purified by silica gel column chromatography (hexane–ethyl acetate, 20:1 to 1:1, stepwise). The active fraction was further purified by preparative reversed-phase MPLC (methanol–water gradient), Sephadex LH-20 chromatography (chloroform–methanol, 1:1), and final MPLC (50% aqueous methanol to 100% methanol), yielding 14 mg of the active compound. Lipopolysaccharide (LPS; #L2630) and dexamethasone (DEX; #D4902) were purchased from Sigma-Aldrich (St. Louis, MO, USA). To evaluate the anti-inflammatory effects of araliadiol, RAW 264.7 macrophage cells were obtained from the Korean Cell Line Bank (Seoul, Republic of Korea). Cells were cultured in RPMI-1640 medium (L0498; Biowest, Nuaillé, France) supplemented with 10% (*v*/*v*) fetal bovine serum (FBS; 35-015-CV; Corning, NY, USA) and maintained at 37 °C in a humidified atmosphere containing 5% CO_2_.

### 2.2. Intracellular Adenosine Triphosphate (ATP) Detection Assay

An intracellular ATP detection assay was conducted to assess the viability of RAW 264.7 cells. Cells (1 × 10^4^) were seeded into 96-well plates and incubated at 37 °C for 24 h. Thereafter, the cells were treated with varying concentrations of araliadiol (0–1 μg/mL) or DEX (1 μg/mL), in the presence or absence of LPS (1 μg/mL) for up to 48 h. Following treatment, ATP levels were measured using a CellTiter-Glo^®^ 2.0 Cell Viability Assay kit (#G9242; Promega, Madison, WI, USA). The supernatant in each well was discarded, and the cells were washed once with PBS. Then, 100 μL of Dulbecco’s phosphate-buffered saline (DPBS; LB001-02; Welgene, Gyeongsan-si, Republic of Korea) was added to each well, followed by the addition of 100 μL of the reagent. The plates were shaken on an orbital shaker for 2 min to induce cell lysis. Subsequently, the plates were incubated at 25 °C in the dark for 10 min to stabilize the luminescent signal. The relative luminescence unit was then measured using a Synergy™ HTX Multi-Mode Microplate Reader (Bioteck, Winooski, VT, USA).

### 2.3. Crystal Violet Staining Assay

To evaluate the proliferative effects of araliadiol on RAW 264.7 cells, a crystal violet staining assay was performed. Cells (1 × 10^5^) were seeded into 12-well plates and maintained at 37 °C for 24 h. The cells were then treated with various concentrations of araliadiol (0–1 μg/mL) or DEX (1 μg/mL), in the presence or absence of LPS (1 μg/mL) for 48 h. After treatment, the cells were washed twice with 300 μL of DPBS, and any remaining buffer was aspirated to allow complete drying. Each well was stained for 1 h using a 0.5% crystal violet solution (#6408; BIOPURE, Namyangju-si, Republic of Korea) prepared in 20% methanol. Excess staining was removed by washing the wells five times with distilled water, and the plates were air-dried. The retained dye was then solubilized using 100% methanol, and cell proliferation was quantified by measuring the absorbance at 570 nm using a microplate reader.

### 2.4. Cell Fractionation and Western Blot Analysis

RAW 264.7 cells (8 × 10^5^) were seeded into 60 mm dishes and incubated at 37 °C for 24 h. The cells were then treated with various concentrations of araliadiol (0–1 µg/mL) or DEX (1 µg/mL), in the presence or absence of LPS (1 µg/mL) for 24 h. Total cell lysates were prepared using a radio-immunoprecipitation assay (RIPA) lysis buffer, while nuclear and cytoplasmic fractions were isolated using NE-PER™ Nuclear and Cytoplasmic Extraction Reagents (#78833; Thermo Fisher Scientific, Waltham, MA, USA). All lysis and extraction buffers were supplemented with a protease inhibitor cocktail (#04693116001; Roche; Merck KGaA, Darmstadt, Germany) and a phosphatase inhibitor cocktail (#4906845001; Roche; Merck KGaA). Protein concentrations were determined using a Pierce™ BCA Protein Assay Kit (#23225; Thermo Fisher Scientific). Western blot analysis was performed following previously described protocols [30]. Details of the primary antibodies and their respective dilutions are provided in Table 1.

### 2.5. Immunofluorescence Staining

RAW 264.7 cells (3 × 10^5^) were seeded into 6-well plates and incubated at 37 °C for 24 h. The cells were then treated with various concentrations of araliadiol (0–1 µg/mL) or DEX (1 µg/mL), in the presence or absence of LPS (1 µg/mL) for 24 h. Following treatment, the cells were fixed with 4% paraformaldehyde (#F8775; Sigma-Aldrich) for 15 min at room temperature and washed three times with DPBS for 5 min each. The cells were then permeabilized with 0.5% Triton X-100 (#9002-03-1; BIOPURE) and blocked with a solution containing 10% normal goat serum (#005-000-121; Jackson ImmunoResearch, West Grove, PA, USA), 2% Tween-20 (#9005-64-5; BIOPURE), and 1% bovine serum albumin (#BSA025; Bovogen, Melbourne, VIC, Australia). The samples were incubated overnight at 4 °C, with primary antibodies against Cleaved caspase-3, Cox-2, p65, and Stat1, each diluted 1:200. The following day, secondary antibodies―Alexa Fluor^®^ 488 Goat anti-Rabbit IgG (#A-11008; Invitrogen; Thermo Fisher Scientific, Carlsbad, CA, USA) and Alexa Fluor^®^ 568 Goat anti-Rabbit IgG (#A-11011; Invitrogen; Thermo Fisher Scientific)―were applied at a 1:500 dilution and incubated for 1 h at 37 °C. Immunofluorescence images were acquired using a Zeiss Axiovert 200 M fluorescence microscope (Carl Zeiss AG, Baden-Württemberg, Germany). Depending on the fluorophores used, appropriate FITC, Rhodamine, or DAPI filters were utilized.

### 2.6. Quantitative Reverse Transcriptase–Polymerase Chain Reaction (qRT-PCR)

RAW 264.7 cells (8 × 10^5^) were seeded into 60 mm dishes and incubated at 37 °C for 24 h. The cells were then treated with various concentrations of araliadiol (0–1 µg/mL) or DEX (1 µg/mL), in the presence or absence of LPS (1 µg/mL) for up to 24 h. The total RNA was extracted using the RiboEx™ reagent (#301-001; Geneall Biotechnology, Seoul, Republic of Korea), and 1000 ng of RNA was reverse-transcribed into complementary DNA (cDNA) using M-MLV reverse transcriptase (#28025021; Thermo Fisher Scientific). Quantitative PCR amplification of target genes was performed using HOT FIREPol^®^ EvaGreen^®^ qPCR Mix Plus (#08-24-0000; SOLIS BIODYNE, Tartu, Estonia) in conjunction with the StepOnePlus Real-Time PCR System (Thermo Fisher Scientific). qRT-PCR was conducted according to previously described protocols [31]. The relative mRNA expression levels were calculated using the 2^−ΔΔCT^ method, with *β-actin* used as the internal control for normalization. The primer sequences utilized in this study are listed in Table 2.

### 2.7. Quantitation of Nitric Oxide (NO) Production

RAW 264.7 cells (1 × 10^5^) were seeded into 96-well plates and incubated for 24 h. Following incubation, the cells were treated with various concentrations of araliadiol (0–1 µg/mL) or DEX (1 µg/mL), in the presence or absence of LPS (1 µg/mL) for 24 h. After treatment, the culture supernatants were collected and mixed with the Griess reagent (#G4410; Sigma-Aldrich) in a 1:1 ratio, followed by incubation at room temperature for 15 min. NO production was quantified by measuring the absorbance at 540 nm using a microplate reader and was calculated based on a standard curve generated using sodium nitrite (#237213; Sigma-Aldrich).

### 2.8. Enzyme-Linked Immunosorbent Assay (ELISA)

RAW 264.7 cells (8 × 10^5^) were seeded into 60 mm dishes and incubated at 37 °C for 24 h. The cells were then treated with various concentrations of araliadiol (0–1 µg/mL) or DEX (1 µg/mL), in the presence or absence of LPS (1 µg/mL) for up to 24 h. Following treatment, the culture supernatants were collected and centrifuged at 12,000 rpm for 5 min. The resulting supernatants were analyzed using a PGE_2_ ELISA kit (ADI-900-001; ENZO Life Sciences, Farmingdale, NY, USA) to quantify prostaglandin E_2_ secretion, according to the manufacturer’s instructions.

### 2.9. Statistical Analysis

All statistical analyses were performed based on a minimum of three independent experiments. One-way analysis of variance (ANOVA) was used to assess statistical differences among groups, employing the GraphPad Prism software (version 8.0.1, San Diego, CA, USA). When significant differences were identified, Tukey’s post hoc test was applied for multiple comparisons within each treatment group. Data are expressed as the mean ± standard deviation (SD), and statistical significance was defined as a *p*-value of less than 0.05.

## 3. Results

### 3.1. Araliadiol Attenuates Cytotoxicity and Cell Death in LPS-Induced Hyperinflammatory Responses in RAW 264.7 Cells

Cell death is a physiological process essential for maintaining tissue homeostasis and facilitating recovery following acute injury in inflammatory environments. However, excessive or dysregulated cell death contributes to the pathogenesis of inflammatory diseases and is increasingly recognized as a driver of chronic inflammation [32]. Intracellular contents released from dying cells activate innate immune responses, particularly lytic forms of cell death such as necroptosis and pyroptosis, and lead to the release of numerous proinflammatory mediators, including damage-associated molecular patterns (DAMPs), which can amplify hyperinflammatory responses [32,33]. Recent studies have highlighted the therapeutic value of suppressing excessive cell death in chronic inflammatory conditions. For example, animal models have shown that the overexpression of BCL-2 or caspase inhibition to prevent apoptosis improves survival in sepsis [34]. Similarly, the genetic deletion of inhibitors of apoptosis proteins (IAPs), including cIAP1, cIAP2, and xIAP, has been associated with the development of inflammatory skin diseases in mice [35]. Moreover, elevated levels of apoptosis and necrosis have been shown to exacerbate inflammatory disorders such as IBD, psoriasis, eczema, and RA [33]. Based on this evidence, we evaluated the protective effects of araliadiol against LPS-induced cytotoxicity and cell death in RAW 264.7 cells. Initially, ATP content and crystal violet staining assays determined that 1 µg/mL of LPS induces significant cytotoxicity in RAW 264.7 cells (Appendix A); therefore, this concentration was used in subsequent experiments.

An ATP-based cell viability assay was used to determine a non-toxic working concentration of araliadiol. As shown in Figure 1a, treatment with araliadiol for 48 h did not induce significant cytotoxicity at concentrations up to 1 µg/mL. Therefore, concentrations below this threshold were used in all subsequent experiments. DEX, a synthetic glucocorticoid, was included as a positive control. Figure 1b,c shows the cytoprotective effects of araliadiol against LPS-induced toxicity as determined by ATP quantification and crystal violet staining, respectively. As shown in Figure 1b, LPS treatment significantly reduced cell viability to 51.870%. However, co-treatment with araliadiol at 0.25, 0.5, and 1 µg/mL increased viability to 59.611%, 59.922%, and 61.994%, respectively. DEX co-treatment restored cell viability to 74.726%. Similarly, as seen in Figure 1c, crystal violet staining revealed reduced dye uptake in the LPS-treated group (11.679%), which was significantly restored with araliadiol co-treatment to 13.637%, 21.986%, and 25.099%. DEX (1 µg/mL) again exhibited a strong protective effect, increasing dye uptake to 51.643%.

To assess whether these protective effects involved the modulation of cell death-related pathways, the expression of key markers was examined by Western blotting (Figure 1d). Araliadiol (1 µg/mL) co-treatment restored Bcl-2 expression, which was reduced by LPS, increasing it from 0.63-fold to 0.97-fold. The expression of Cleaved caspase-3 and Cleaved PARP, markers of apoptosis, was markedly reduced in a concentration-dependent manner. In LPS-treated cells, Cleaved caspase-3 and Cleaved PARP levels increased by 19.19-fold and 3.53-fold, respectively, and were reduced to 5.57-fold and 1.27-fold following co-treatment with araliadiol at 1 µg/mL. Immunofluorescence staining for Cleaved caspase-3 corroborated the Western blot findings (Figure 1e). LPS-treated cells showed strong cytoplasmic and nuclear fluorescence signals, indicating elevated cell death, which were markedly reduced in araliadiol co-treated cells (0.25–1 µg/mL).

Collectively, these findings indicate that araliadiol reduces LPS-induced cytotoxicity and cell death in RAW 264.7 cells and suggest its potential as a cytoprotective agent.

### 3.2. Araliadiol Downregulates the Expression of Pro-Inflammatory Mediators, Highlighting Its Therapeutic Potential in Inflammatory Diseases

Inflammatory responses are essential for maintaining tissue homeostasis and are activated by internal and external stimuli, including pathogens and environmental irritants [36]. Although inflammation contributes to the restoration of homeostasis, excessive or uncontrolled activation can lead to tissue damage and is associated with the pathogenesis of various diseases [37]. The inflammatory process involves four principal components: inducers, sensors, mediators, and target tissues [37]. The overproduction of inflammatory mediators, such as amines, peptides, eicosanoids, cytokines, and chemokines, has been implicated in the development of several chronic conditions, including cardiovascular diseases, RA, periodontitis, and IBD [38,39,40,41]. Accordingly, the therapeutic regulation of these mediators remains an area of active investigation. Given the cytoprotective and anti-cell death effects of araliadiol in LPS-induced inflammatory conditions (Figure 1), its role in modulating the expression of pro-inflammatory mediators was evaluated.

In RAW 264.7 cells treated with LPS (1 μg/mL), the gene expression levels of inflammasome-associated markers *Il-1β*, *Nlrp-3*, and *Il-18* increased markedly, by 607.108-, 3.999-, and 9.738-fold, respectively (Figure 2a). Co-treatment with araliadiol (0.25–1 µg/mL) significantly attenuated these elevations in a concentration-dependent manner. Specifically, *Il-1β* expression decreased by 545.870-, 450.918-, and 350.182-fold; *Nlrp-3* by 3.897-, 3.371-, and 3.237-fold; and *Il-18* by 10.257-, 7.654-, and 6.345-fold with araliadiol at 0.25, 0.5, and 1 μg/mL, respectively. In comparison, co-treatment with DEX (1 μg/mL) reduced *Il-1β*, *Nlrp-3*, and *Il-18* expression by 292.272-, 3.764-, and 4.682-fold, respectively. Notably, araliadiol demonstrated a more pronounced inhibitory effect on *Nlrp-3* expression, with a 0.526-fold greater reduction compared to the DEX group.

Further analysis revealed that araliadiol also reduced the expression of key pro-inflammatory cytokines (Figure 2b). LPS-only treatment increased the expression of *Tnf-α*, *Il-6*, and *Il-12α*, which were suppressed in a concentration-dependent manner by araliadiol. Specifically, *Tnf-α* expression decreased from 10.663-fold (LPS only) to 10.405-, 8.680-, and 7.528-fold with 0.25, 0.5, and 1 μg/mL araliadiol, respectively. *Il-6* expression, which rose to 61,465.193-fold following LPS treatment, was reduced by 57,457.778-, 45,162.079-, and 34,788.469-fold. *Il-12α* expression declined from 18.893-fold to 10.621-, 7.166-, and 7.091-fold following araliadiol co-treatment. Araliadiol was more effective than DEX (1 μg/mL) in reducing *Il-12α* levels, achieving a 17.375-fold greater reduction at equivalent concentrations.

Figure 2c presents the effects of araliadiol on pro-inflammatory chemokines. In the LPS-only group, *Ccl17*, *Ccl23*, and *Cxcl9* expression increased by 4.040-, 6.764-, and 2.089-fold, respectively. Co-treatment with araliadiol (0.25, 0.5, and 1 μg/mL) reduced *Ccl17* expression by 1.931-, 1.757-, and 1.615-fold; *Ccl23* by 5.469-, 5.567-, and 5.148-fold; and *Cxcl9* by 1.479-, 1.514-, and 1.122-fold, respectively. DEX co-treatment showed comparable effects, reducing expression by 1.483-, 1.667-, and 1.122-fold for *Ccl17*, *Ccl23*, and *Cxcl9*, respectively, supporting the anti-inflammatory activity of araliadiol.

Figure 2d,e illustrates the impact of araliadiol on *iNos* expression and NO production, both key indicators of inflammatory activation. LPS treatment elevated *iNos* expression by 13.782-fold, which was reduced to 10.345-, 9.307-, and 8.278-fold with 0.25, 0.5, and 1 μg/mL araliadiol, respectively. Correspondingly, NO production decreased from 55.034-fold (LPS only) to 49.235-, 43.101-, and 29.908-fold. Araliadiol (1 μg/mL) reduced NO production by 5.966-fold more than DEX (1 μg/mL), further supporting its potent anti-inflammatory activity.

Collectively, the data presented in Figure 2 indicate that araliadiol significantly downregulates the expression of multiple inflammatory mediators in LPS-stimulated RAW 264.7 cells. These findings support the pharmacological potential of araliadiol as a therapeutic agent in inflammatory diseases.

### 3.3. Araliadiol Has Therapeutic Potential for Alleviating Inflammatory Hyperalgesia by Downregulating Cox-2 and PGE_2_ Expression

Pain, a primary symptom of inflammation, is one of four classical inflammatory responses—redness, heat, swelling, and pain. It serves as an essential early warning signal initiated by nociceptors located at the peripheral terminals of sensory neurons [42]. These nociceptors contain key proteins, including transient receptor potential vanilloid 1 (TRPV1) and voltage-gated sodium channels (Nav), which detect noxious stimuli and convert them into electrical signals perceived as pain [43]. Acute pain functions as a protective mechanism, preventing harm and promoting healing, whereas chronic or severe pain may impair daily functioning, elevate mortality risk, and substantially diminish quality of life (QoL), often contributing to psychological distress [44].

Hyperalgesia, or increased sensitivity to pain, commonly results from the excessive release of inflammatory mediators comprising the “inflammatory soup”, which includes purines, amines, growth factors, cytokines, chemokines, and prostaglandins [45]. These mediators act via G protein-coupled receptor (GPCR) signaling pathways to upregulate TRPV1 and Nav expression on sensory neurons, thereby reducing nociceptor activation thresholds and facilitating pain transmission [46]. Inflammatory signaling is a key driver of hyperalgesia, converting normally non-painful stimuli into painful sensations [45].

Targeting cyclooxygenase-2 (Cox-2) and its downstream product prostaglandin E_2_ (PGE_2_) is a widely adopted strategy for managing inflammatory pain. NSAIDs exert analgesic effects by inhibiting Cox enzymes, thereby reducing PGE_2_ production [47]. Previous studies have shown that the pharmacological inhibition or genetic ablation of Cox-2 effectively reduces pain hypersensitivity [48,49]. Similarly, the inhibition of microsomal prostaglandin E synthase-1 (mPGES-1) [50], an enzyme involved in PGE_2_ biosynthesis, and the blockade of prostaglandin E_2_ receptor 4 (EP4) alleviated inflammatory joint pain in animal models [51]. Given previous findings that araliadiol suppresses multiple inflammatory mediators, including inflammasome-related genes, cytokines, and chemokines (Figure 2), we evaluated whether araliadiol could also modulate Cox-2 expression and thereby attenuate PGE_2_ production.

The initial analysis of PGE_2_ levels, a key biomarker of inflammatory pain, revealed a substantial increase in the LPS-only treated group, reaching 444.554-fold relative to the solvent-treated control. Co-treatment with araliadiol at 0.25, 0.5, and 1 μg/mL significantly reduced PGE_2_ levels by 372.432-, 349.594-, and 239.353-fold, respectively (Figure 3a). In comparison, co-treatment with DEX (1 μg/mL) reduced PGE_2_ by 273.321-fold. Notably, araliadiol at 1 μg/mL exhibited a PGE_2_-suppressing effect approximately 33.967-fold greater than that of DEX at the same concentration.

Considering that Cox-2 is selectively upregulated in inflamed tissues and plays a central role in PGE_2_ biosynthesis [52], we next quantified *Cox-2* mRNA levels. In the LPS-only treated group, *Cox-2* expression increased by 229.524-fold, while co-treatment with araliadiol at 0.25, 0.5, and 1 μg/mL reduced expression by 179.706-, 150.125-, and 137.410-fold, respectively (Figure 3b). A similar trend was observed at the protein level: Cox-2 expression increased 38.11-fold in the LPS-only group and was reduced by 26.01-fold with araliadiol (1 μg/mL) co-treatment (Figure 3c). Co-treatment with DEX (1 μg/mL) reduced Cox-2 mRNA and protein expression by 64.163- and 20.31-fold, respectively. Immunofluorescence staining further confirmed these findings by visualizing Cox-2 protein expression (Figure 3d). High fluorescence intensity was observed in the LPS-only group, while markedly lower intensity was seen in cells co-treated with araliadiol or DEX, indicating reduced Cox-2 expression.

These results collectively indicate that araliadiol reduces PGE_2_ levels by downregulating *COX2* gene and protein expression. Given the pivotal role of PGE_2_ in inflammatory pain signaling, these findings support the therapeutic potential of araliadiol for alleviating inflammatory hyperalgesia.

### 3.4. Araliadiol Suppresses Nfκb and Stat1 Signaling Pathways in LPS-Stimulated RAW 264.7 Cells

Transcription factors regulate gene expression by binding to specific DNA sequences (motifs), functioning as master regulators within cells and modulating essential physiological processes, including differentiation, proliferation, aging, and apoptosis [53]. In the context of inflammation, certain transcription factors are particularly critical. NFκB activation induces the expression of inflammatory mediators such as TNF-α, IL-1β, IL-6, and IL-8 [54,55], whereas the STAT1 pathway promotes the transcription of inflammatory genes such as iNOS, COX, vascular cell adhesion molecules (VCAMs), and intercellular adhesion molecules (ICAMs), thereby amplifying the inflammatory response [56]. Previous studies have linked aberrant NFκB pathway activation to chronic inflammatory conditions, including asthma, atherosclerosis, RA, and IBD [57]. Similarly, enhanced STAT1 signaling has been associated with pathologies such as ischemia/reperfusion injury, unstable angina, celiac disease, and psoriasis [56,58]. Accordingly, the inhibition of NFκB and STAT1 is a strategic focus in the development of anti-inflammatory therapeutics. Natural compounds such as berbamine from *Berberis vulgaris* [59] and curcumin from turmeric have shown efficacy in mitigating inflammatory diseases by modulating these transcriptional pathways [60]. Given the critical roles of NFκB and STAT1 in inflammatory regulation, the present study investigated whether araliadiol suppresses inflammatory signaling by targeting these transcription factors (Figure 4 and Figure 5).

As shown in Figure 4a, treatment with LPS alone markedly increased the phosphorylation of IκB-α at serine 32 and p65 at serine 536, which are key steps in Nfκb activation. Co-treatment with araliadiol (0.25–1 µg/mL) reduced the phosphorylation of p65 (Ser536) in a concentration-dependent manner, with greater efficacy than DEX (1 μg/mL). Immunofluorescence staining (Figure 4b) confirmed that araliadiol (1 μg/mL) suppressed the LPS-induced nuclear translocation of p65 and reduced both total and nuclear-localized p65 protein levels. The NFκB/Rel family—comprising RelA (p65), RelB, c-Rel, p50 (p105), and p52 (p100)—activates transcription through nuclear translocation of the p65/p50 heterodimer [61,62]. Figure 4c,d shows that araliadiol inhibited the LPS-induced nuclear translocation of p50 more effectively than DEX, achieving a 0.395-fold greater reduction.

The effects of araliadiol on Stat1 activation were then evaluated (Figure 5). LPS treatment significantly increased total Stat1 protein levels and phosphorylation at tyrosine 701 and serine 727, modifications necessary for transcriptional activation. Co-treatment with araliadiol (0.25–1 µg/mL) substantially reduced overall Stat1 expression and significantly decreased phosphorylation at serine 727 in a concentration-dependent manner. At 1 μg/mL, araliadiol suppressed serine 727 phosphorylation more effectively than DEX (1 μg/mL). Although no significant effect was observed on tyrosine 701 phosphorylation, additional analyses (Figure 5b–d) confirmed that araliadiol reduced STAT1 transcriptional activity. Both immunofluorescence and nuclear fractionation demonstrated a dose-dependent reduction in total and nuclear-localized Stat1 levels following treatment.

Together, these results indicate that araliadiol suppresses the LPS-induced activation of Nfκb and the Stat1 signaling in RAW 264.7 cells. By targeting these transcription factors, araliadiol may confer anti-inflammatory and anticytotoxic effects and downregulate hyperalgesia-associated biomarkers, thereby supporting its potential utility in the management of inflammation.

## 4. Discussion

Inflammatory disorders encompass a broad spectrum of diseases that substantially impact human health. According to the World Health Organization (WHO), inflammatory conditions are among the leading causes of mortality worldwide, with three out of five individuals dying from chronic inflammatory diseases [63]. Individuals with immune-mediated inflammatory conditions often experience diminished QoL [64,65]. Coupled with high socioeconomic burden, including medication costs, the loss of productivity, and work-related absenteeism, there remains a critical need to develop anti-inflammatory agents that are safe, effective, and economically viable [66]. Conventional synthetic anti-inflammatory drugs are frequently associated with adverse effects, whereas biological therapies are often cost-prohibitive. Consequently, there is considerable interest in natural anti-inflammatory compounds that are both affordable and biocompatible [67,68,69]. Therefore, we evaluated the pharmacological effects of araliadiol, a phytochemical isolated from *C. asiatica*, as a potential anti-inflammatory agent. Araliadiol attenuated cytotoxicity, suppressed inflammatory responses, and downregulated the expression of inflammation-associated pain mediators in LPS-stimulated RAW 264.7 cells. Notably, at an equivalent concentration, araliadiol demonstrated anti-inflammatory efficacy comparable to the positive control, DEX. These effects may be mediated through the inhibition of the Nfκb and Stat1 signaling pathways.

Araliadiol markedly reduced cytotoxicity in LPS-treated RAW 264.7 cells, as evidenced by increased Bcl-2 expression and decreased levels of Cleaved caspase-3 and Cleaved PARP-1 (Figure 1). Previous reports have proposed that the inhibition of apoptosis may offer therapeutic benefits in sepsis, a systemic inflammatory syndrome triggered by severe infection. In this context, Bcl-2 overexpression in immune cells has been shown to protect against sepsis-associated apoptosis, improving survival by up to threefold compared to controls [70]. Similarly, the pharmacological inhibition or genetic deletion of caspase-3 significantly enhanced survival in sepsis models [70]. In caspase-3 knockout mice with NASH and liver fibrosis, reduced cell death and markedly lower cytokine expression have been observed [71]. In RA, caspase-3 activation is reported in monocytes and macrophages [72], and the dual inhibition of caspase-3 and caspase-8 has alleviated osteoarthritis symptoms in animal models [72]. Given the role of excessive cell death in the pathogenesis of inflammatory diseases, the anti-cytotoxic activity of araliadiol suggests its potential utility in conditions such as sepsis and arthritis [32,72].

Furthermore, araliadiol reduced the expression of various pro-inflammatory mediators in LPS-stimulated RAW 264.7 cells. Specifically, it significantly downregulated ten genes involved in inflammasome signaling, cytokine and chemokine production, and nitric oxide synthesis (Figure 2). IL-1β, generated through inflammasome activation, plays a key role in amplifying immune responses and is implicated in the pathogenesis of several immune disorders [73]. Canakinumab, a monoclonal antibody targeting IL-1β, has been approved by the FDA for treating autoimmune conditions such as familial cold autoinflammatory syndrome and Muckle–Wells syndrome [74]. Similarly, anti-TNF therapies including etanercept, infliximab, and adalimumab are commonly used in the treatment of autoimmune diseases such as rheumatoid arthritis, Crohn’s disease, psoriatic arthritis, and psoriasis [75,76]. Anti-TNF IgG has also been shown to ameliorate symptoms of collagen-induced arthritis in murine models [77,78]. Additionally, neutralizing antibodies against cytokines such as TNF-α, IL-6, and IL-12 have demonstrated efficacy in immune-mediated conditions, including Crohn’s disease [79,80,81].

Chemokines such as CCL17, CCL23, and CXCL9 are frequently upregulated in autoimmune diseases [82,83,84]. GSK3858279, an anti-CCL17 monoclonal antibody, has shown efficacy in relieving osteoarthritis-associated pain [85], and CXCL9 inhibition has demonstrated therapeutic potential in ulcerative colitis [86]. NO, a pro-inflammatory free radical, is a key biomarker in conditions such as RA, osteoarthritis, ankylosing spondylitis, and IBD [87]. Use of NOS inhibitors in murine arthritis models significantly reduces inflammation [88]. Given the therapeutic importance of suppressing inflammatory mediators such as TNF-α, IL-1β, and IL-6, the anti-inflammatory activity of araliadiol—particularly the substantial downregulation (by 26,676.724-fold) of *Il-6* expression at 1 μg/mL—supports its potential as a candidate anti-inflammatory agent for a range of inflammatory diseases [89].

Figure 3 presents the inhibitory effects of araliadiol on markers associated with inflammatory pain. Chronic inflammation contributes to long-term alterations in the sensory system that underlie persistent pain, emphasizing the importance of effective pain management for improving QoL [90]. Pain hypersensitivity arises when peripheral sensory neurons respond to inflammatory mediators, resulting in elevated TRPV1 and Nav receptor levels and reduced nociceptor activation thresholds [46]. PGE_2_ is a principal mediator of inflammatory pain hypersensitivity and a critical target in pain management strategies [47]. Previous studies have shown that the use of a PGE_2_ receptor antagonist (AH23848) or EP4 knockdown via short hairpin RNA (shRNA) alleviated inflammatory pain in mice [91]. Clinical data have also demonstrated that the EP1 antagonist ZD6416 reduces esophageal hyperalgesia [92]. In our study, araliadiol (1 μg/mL) significantly reduced LPS-induced Cox-2 protein expression by 12.099-fold and PGE_2_ production by 205.201-fold. Given that common pain-relieving drugs such as aspirin, naproxen, and celecoxib modulate pain by inhibiting Cox-2 and reducing prostaglandin levels, the reduction in Cox-2 expression and PGE_2_ production by araliadiol highlights its potential as a natural anti-pain agent [93].

This study employed the LPS-induced RAW 264.7 cell model, a well-established in vitro model of inflammation, with DEX as a positive control [94]. A persistent challenge in the development of natural product-based drugs is the need for relatively high concentrations compared to synthetic agents. For example, natural compounds such as capsaicin (10–50 μM) [95], berberine (10–100 μM) [96], EGCG (10.9–87.2 μM) [97], and quercetin (10–100 μM) [98] require comparatively high concentrations to achieve anti-inflammatory efficacy. Similarly, acetylsalicylic acid and sodium salicylate exhibit anti-inflammatory effects at concentrations of 2–10 mM [99]. In contrast, araliadiol showed potent anti-inflammatory effects at substantially lower concentrations (0.25–1 μg/mL, approximately 1–4.3 μM), indicating notable potency despite being a phytochemical. Furthermore, when tested at the same concentration as DEX (1 μg/mL), araliadiol exhibited comparable intracellular anti-inflammatory effects, with greater suppression of *Nlrp3* and *Il-12α* expression and more pronounced reductions in NO and PGE_2_ secretion.

To the best of our knowledge, this is the first report to comprehensively characterize the anti-inflammatory effects of araliadiol using biochemical analysis. Despite being a naturally derived compound, araliadiol exhibited anti-inflammatory efficacy comparable to that of the synthetic drug dexamethasone at the same low concentration (1 µg/mL). While our findings highlight its potential as a novel therapeutic candidate for inflammatory diseases, further validation in in vivo models and clinical trials is warranted. Given that this study demonstrated the pharmacological potential of araliadiol in an in vitro model, additional pharmacokinetic and pharmacodynamic investigations are required to facilitate clinical translation. These should include regression analyses of in vitro cytotoxicity data (e.g., IC_50_) and in vivo toxicity assessments (e.g., LD_50_ in animal models). Future studies should also evaluate the disease-specific efficacy of araliadiol under pathological conditions such as psoriasis and arthritis.

## 5. Conclusions

In summary, our findings demonstrate that araliadiol exerts anticytotoxic and anti-inflammatory effects and suppresses inflammatory pain mediators in LPS-stimulated RAW 264.7 cells. Araliadiol reduced LPS-induced cytotoxicity and cell death, notably by upregulating Bcl-2 expression while downregulating cell death markers such as Cleaved caspase-3 and Cleaved PARP. Moreover, araliadiol effectively alleviated LPS-induced inflammation by significantly suppressing the expression of inflammasome-related markers (*Il-1β*, *Nlrp3*, and *Il-18*), proinflammatory cytokines (*Tnf-α*, *Il-6*, and *Il-12α*), chemokines (*Ccl17*, *Ccl23*, and *Cxcl9*), and nitric oxide. Additionally, araliadiol inhibited the expression of the hyperalgesic marker Cox-2 and reduced PGE_2_ production. These anti-inflammatory and analgesic effects may be mediated through the suppression of Nfκb and Stat1 signaling pathways. These results support its pharmacological potential as a novel anti-inflammatory agent. Furthermore, araliadiol exhibited anti-inflammatory efficacy comparable to that of DEX at the same concentration, indicating its potential as a safe and effective therapeutic candidate derived from natural products.

## Figures and Tables

**Figure 1 biomedicines-13-01408-f001:**
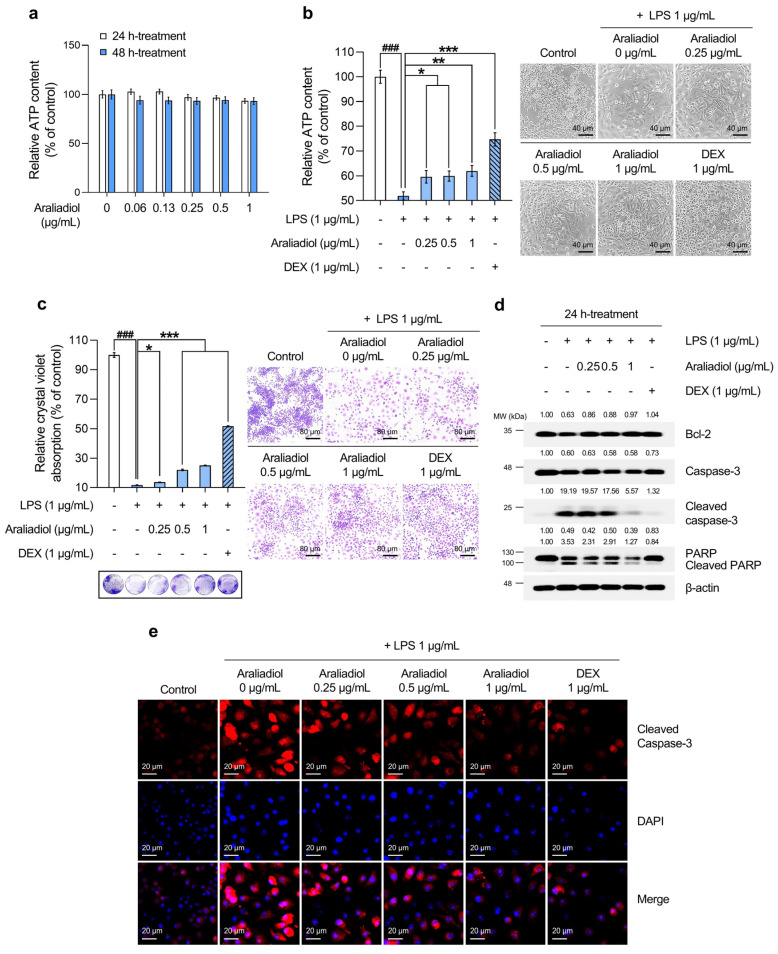
The protective effects of araliadiol against LPS-induced cytotoxicity and cell death in RAW 264.7 cells. (**a**,**b**) RAW 264.7 cells were seeded into 96-well plates (1 × 10^4^ cells/well) and incubated for 24 h. The cells were then treated with araliadiol (0–1 µg/mL) or DEX (1 µg/mL), in the presence or absence of LPS (1 µg/mL) for up to 48 h. Cell viability was determined using an ATP-content assay. Representative images were captured using an inverted phase-contrast microscope (80× magnification; scale bar: 40 µm). (**c**) Cells were seeded into 12-well plates (1 × 10^5^ cells/well) and incubated for 24 h. Following treatment with the indicated chemicals for 48 h, cytotoxicity was evaluated using a crystal violet staining assay. Representative images were acquired under an inverted phase-contrast microscope (40× magnification; scale bar: 80 µm). (**d**) Cells were seeded into 60 mm dishes (8 × 10^5^ cells/dish) and treated with the respective compounds for 24 h. Protein expression levels of cell death-associated markers (Bcl-2, Caspase-3, Cleaved caspase-3, and Cleaved PARP-1) were analyzed via Western blotting. β-actin was used as a loading control, and protein band intensities were quantified using the ImageJ software (version 1.53t). (**e**) The cells were seeded into 6-well plates (3 × 10^5^ cells/well) and treated with the indicated chemicals for 24 h. The sub-cellular localization of Cleaved caspase-3 was examined via immunofluorescence staining. Representative images were captured using an Axiovert 200 M inverted fluorescence microscope. Cleaved caspase-3 was visualized using a Rhodamine filter, while nuclei were visualized using a DAPI filter (160× magnification; scale bar: 20 µm). Data are presented as the mean ± SD from three independent experiments. Statistical analysis was performed using one-way ANOVA followed by Tukey’s post hoc test. LPS, lipopolysaccharide; DEX, dexamethasone; ATP; adenosine triphosphate; Bcl-2, B-cell leukemia/lymphoma 2 protein; PARP-1, poly(ADP-ribose) polymerase 1. ### *p* < 0.001 compared with the solvent-treated vehicular control group. * *p* < 0.05; ** *p* < 0.01; *** *p* < 0.001 compared with the LPS-treated negative control group.

**Figure 2 biomedicines-13-01408-f002:**
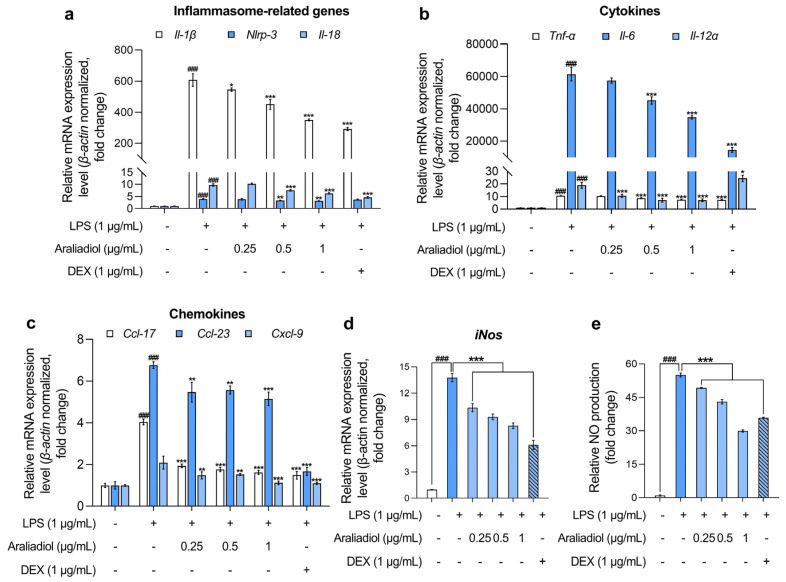
The anti-inflammatory effects of araliadiol in LPS-stimulated RAW 264.7 cells. (**a**–**d**) RAW 264.7 cells were seeded into 60 mm dishes (8 × 10^5^ cells/dish) and incubated for 24 h. The cells were subsequently treated with araliadiol (0–1 µg/mL) or DEX (1 µg/mL), in the presence or absence of LPS (1 µg/mL) for up to 24 h. mRNA expression levels of inflammasome-related genes (*Il-β*, *Nlrp-3*, and *Il-18*) (**a**), cytokines (*Tnf-α*, *Il-6*, and *Il-12 α*) (**b**), chemokines (*Ccl-17*, *Ccl-23*, and *Cxcl-9*) (**c**), and *iNos* (**d**) were analyzed via qRT-PCR, with the expression normalized to *β-actin*. (**e**) The cells were seeded into 96-well plates (1 × 10^5^ cells/well) and incubated for 24 h. Following treatment with the indicated chemicals for 24 h, the supernatants from each well were collected, and NO production was quantified using the Griess assay. Data are presented as the mean ± SD from three independent experiments. Statistical analysis was performed using one-way ANOVA followed by Tukey’s post hoc test. Il, interleukin; Nlrp, nucleotide-binding oligomerization domain-like receptor pyrin domain containing 3; Tnf, tumor necrosis factor; Ccl, chemokine C-C motif ligand; Cxcl, C-X-C motif chemokine ligand; iNos, inducible nitric oxide synthase; qRT-PCR, quantitative reverse transcription polymerase chain reaction. ### *p* < 0.001 compared with the solvent-treated vehicular control group. * *p* < 0.05; ** *p* < 0.01; *** *p* < 0.001 compared with the LPS-treated negative control group.

**Figure 3 biomedicines-13-01408-f003:**
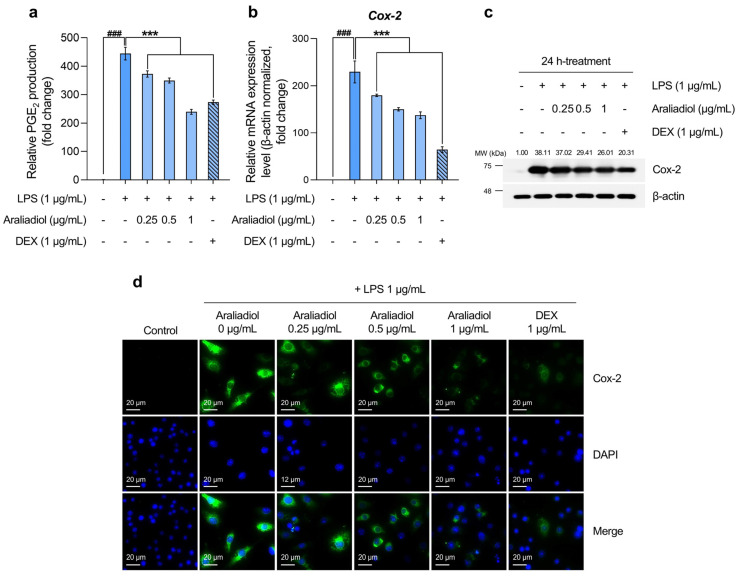
The downregulatory effects of araliadiol on markers associated with inflammatory pain hypersensitivity. (**a**,**b**) RAW 264.7 cells were seeded into 60 mm dishes (8 × 10^5^ cells/dish) and incubated for 24 h. Cells were then treated with araliadiol (0–1 µg/mL) or DEX (1 µg/mL), in the presence or absence of LPS (1 µg/mL) for 24 h. PGE_2_ production was quantified using an ELISA kit (**a**), and *Cox-2* mRNA expression levels were assessed by qRT-PCR and normalized to *β-actin* expression (**b**). (**c**) The cells were seeded into 60 mm dishes (8 × 10^5^ cells/dish) and treated with the designated chemicals for 24 h. The protein expression of Cox-2 was analyzed by Western blotting. β-actin served as a loading control, and band intensities were quantified using ImageJ software (version 1.53t). (**d**) Cells were seeded into 6-well plates (3 × 10^5^ cells/well) and treated with the indicated compounds for 24 h. The sub-cellular localization of Cox-2 was examined via immunofluorescence staining. Representative images were captured using an Axiovert 200 M inverted fluorescence microscope. Cox-2 was visualized using a FITC filter, while nuclei were visualized using a DAPI filter (160× magnification; scale bar: 20 µm). Data are presented as the mean ± SD from three independent experiments. Statistical analysis was performed using one-way ANOVA followed by Tukey’s post hoc test. ELISA, enzyme-linked immunosorbent assay; Cox-2, cyclooxygenase-2; PGE_2_, prostaglandin E_2_. ### *p* < 0.001 compared with the solvent-treated vehicular control group. *** *p* < 0.001 compared with the LPS-treated negative control group.

**Figure 4 biomedicines-13-01408-f004:**
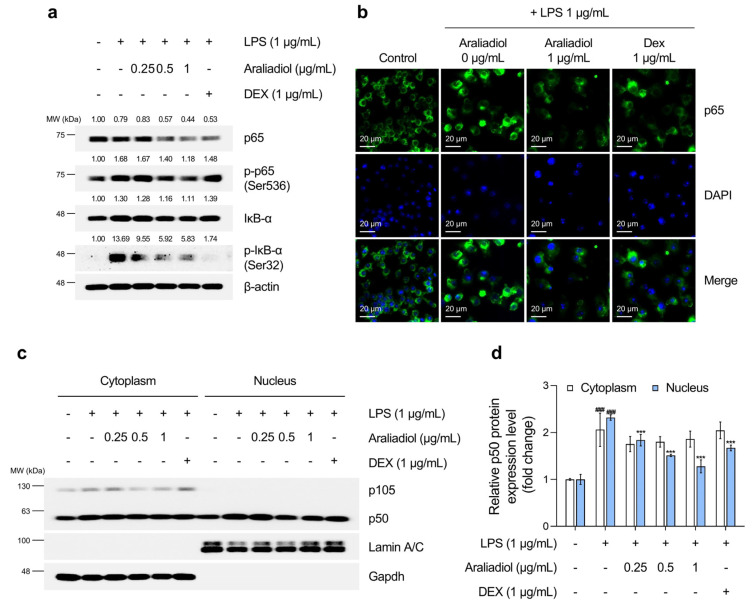
The inhibitory effects of araliadiol on Nfκb signaling in LPS-stimulated RAW 264.7 cells. (**a**) RAW 264.7 cells were seeded into 60 mm dishes (8 × 10^5^ cells/dish) and incubated for 24 h. Cells were then treated with araliadiol (0–1 µg/mL) or DEX (1 µg/mL), in the presence or absence of LPS (1 µg/mL) for 24 h. Nfκb signaling was assessed by Western blotting, with β-actin used as a loading control. (**b**) Cells were seeded into 6-well plates (3 × 10^5^ cells/well) and treated with the indicated chemicals for 24 h. The sub-cellular localization of p65 was examined via immunofluorescence staining. Representative images were captured using an Axiovert 200 M inverted fluorescence microscope. p65 was visualized using a FITC filter, while nuclei were visualized using a DAPI filter (160× magnification; scale bar: 20 µm). (**c**,**d**) Nuclear translocation of p50 was evaluated by Western blotting of nuclear and cytoplasmic fractions. GAPDH and Lamin A/C were used as loading controls for the cytoplasmic and nuclear fractions, respectively. Protein band intensities were quantified using ImageJ software (version 1.53t). Data are presented as mean ± SD from three independent experiments. Statistical analysis was performed using one-way ANOVA followed by Tukey’s post hoc test. Iκb, NF-kappa-B inhibitor; GAPDH, glyceraldehyde-3-phosphate dehydrogenase. ### *p* < 0.001 compared with the solvent-treated vehicular control group. *** *p* < 0.001 compared with the LPS-treated negative control group.

**Figure 5 biomedicines-13-01408-f005:**
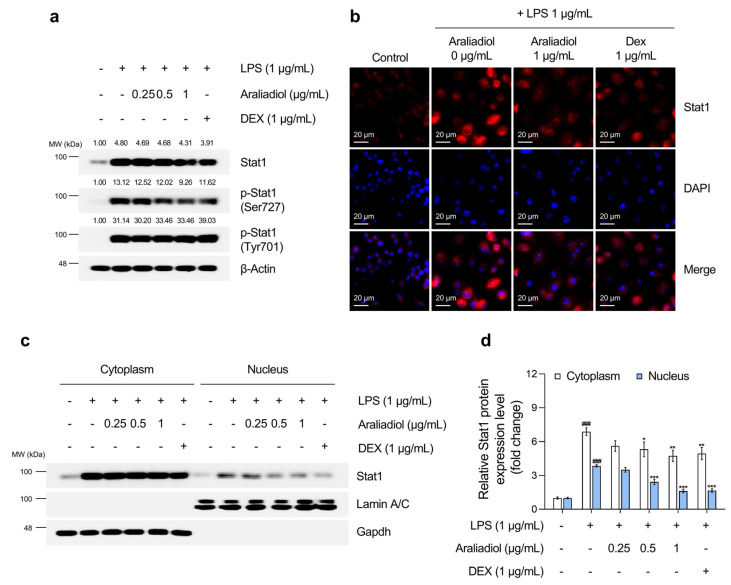
The inhibitory effects of araliadiol on Stat1 signaling in LPS-stimulated RAW 264.7 cells. (**a**) RAW 264.7 cells were seeded into 60 mm dishes (8 × 10^5^ cells/dish) and incubated for 24 h. Cells were then treated with araliadiol (0–1 µg/mL) or DEX (1 µg/mL), in the presence or absence of LPS (1 µg/mL) for 24 h. Stat1 signaling was assessed by Western blotting, with β-actin used as a loading control. (**b**) Cells were seeded into 6-well plates (3 × 10^5^ cells/well) and treated with the indicated chemicals for 24 h. The sub-cellular localization of Stat1 was examined via immunofluorescence staining. Representative images were captured using an Axiovert 200 M inverted fluorescence microscope. Stat1 was visualized using a Rhodamine filter, while nuclei were visualized using a DAPI filter (160× magnification; scale bar: 20 µm). (**c**,**d**) The nuclear translocation of Stat1 was evaluated by Western blotting of nuclear and cytoplasmic fractions. GAPDH and Lamin A/C were used as loading controls for the cytoplasmic and nuclear fractions, respectively. Protein band intensities were quantified using the ImageJ software (version 1.53t). Data are presented as the mean ± SD from three independent experiments. Statistical analysis was performed using one-way ANOVA followed by Tukey’s post hoc test. Stat1, signal transducer and activator of transcription 1. ### *p* < 0.001 compared with the solvent-treated vehicular control group. * *p* < 0.05; ** *p* < 0.01; *** *p* < 0.001 compared with the LPS-treated negative control group.

**Table 1 biomedicines-13-01408-t001:** List of primary antibodies for Western blot analyses.

Antigen	Host	Clonality (Species Reactivity)	Dilution	Manufacturer (Cat. Number)
Bcl-2	Rabbit	Monoclonal (Human, Mouse)	1:1000	CST (#3498)
Caspase-3	Rabbit	Polyclonal (Human, Mouse, Rat, …)	1:1000	CST (#9662)
Cleaved caspase-3	Rabbit	Monoclonal (Human, Mouse, Rat, …)	1:1000	CST (#9664)
PARP	Rabbit	Polyclonal (Human, Mouse, Rat, …)	1:1000	CST (#9542)
β-actin	Mouse	Monoclonal (Human, Mouse, Rat, …)	1:1000	Santa Cruz (#sc-47778)
Cox-2	Rabbit	Monoclonal (Human, Mouse, Rat)	1:1000	CST (#12282)
p65	Rabbit	Monoclonal (Human, Mouse, Rat, …)	1:1000	CST (#8242)
Phospho-p65 (Ser536)	Rabbit	Monoclonal (Human, Mouse, Rat, …)	1:1000	CST (#3033)
Iκb-α	Rabbit	Polyclonal (Human, Mouse, Rat, …)	1:1000	CST (#9242)
Phospho-Iκb-α (Ser32)	Rabbit	Monoclonal (Human, Mouse, Rat, …)	1:1000	CST (#2859)
p50/p105	Rabbit	Monoclonal (Human, Mouse, Rat)	1:1000	CST (#13586)
Lamin A/C	Mouse	Monoclonal (Human, Mouse, Rat, …)	1:1000	CST (#4777)
GAPDH	Rabbit	Monoclonal (Human, Mouse, Rat, …)	1:1000	CST (#5174)
Stat1	Rabbit	Monoclonal (Human, Mouse, Rat, …)	1:1000	CST (#14994)
Phospho-Stat1 (Ser727)	Rabbit	Polyclonal (Human, Mouse, Rat, …)	1:1000	CST (#9177)
Phospho-Stat1 (Tyr701)	Rabbit	Monoclonal (Human, Mouse)	1:1000	CST (#9167)

**Table 2 biomedicines-13-01408-t002:** List of sequences used for qRT-PCR.

Target mRNA	Sequences of Primer	Amplicons (bp)
*β-actin*	F: 5′-GTATGGAATCCTGTGGCATC-3′	322
	R: 5′-AAGCACTTGCGGTGCACGAT-3′	
*Il-1β*	F: 5′-GCCCATCCTCTGTGACTCAT-3′	230
	R: 5′-AGGCCACAGGTATTTTGTCG-3′	
*Il-6*	F: 5′-GAGGATACCACTCCCAACAGACC-3′	141
	R: 5′-AAGTGCATCATCGTTGTTCATACA-3′	
*Tnf-α*	F: 5′-CTACTCCTCAGAGCCCCCAG-3′	231
	R: 5′-TGACCACTCTCCCTTTGCAG-3′	
*Cox-2*	F: 5′-GAAGTCTTTGGTCTGGTGCCTG-3′	133
	R: 5′-GTCTGCTGGTTTGGAATAGTTGC-3′	
*Il-18*	F: 5′-CCATGCTTTCTGGACTCCTGCC-3′	133
	R: 5′-CCATTGTTCCTGGGCCAAGAGG-3′	
*Nlrp3*	F: 5′-GCCCTTGGAGACACAGGACTCA-3′	224
	R: 5′-CCCTGCTGTTTCAGCACCTCAC-3′	
*iNos*	F: 5′-CGAAACGCTTCACTTCCAA-3′	51
	R: 5′-TGAGCCTATATTGCTGTGGCT-3′	
*Cxcl-9*	F: 5′-ATGAAGTCCGCTGTTCTTTTCC-3′	386
	R: 5′-GTCTCTTATGTAGTCTTCCTTG-3′	
*Il-12a*	F: 5′-CGGGACCAAACCAGCACATTGA-3′	105
	R: 5′-GCAGCTCCCTCTTGTTGTGGAA-3′	
*Ccl-17*	F: 5′-ATGCCATCGTGTTTCTGACTGT-3′	99
	R: 5′-GCCTTGGGTTTTTCACCAATC-3′	
*Ccl-22*	F: 5′-AAGCCTGGCGTTGTTTTGAT-3′	99
	R: 5′-TCCCTAGGACAGTTTATGGAGTAGCT-3′	

## Data Availability

The original contributions presented in the study are included in the article; further inquiries can be directed to the corresponding author.

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
