# Peer review of "Pharmacological Evaluation of Araliadiol as a Novel Anti-Inflammatory Agent in LPS-Induced RAW 264.7 Cells"

_biomedicines, 2025, doi:10.3390/biomedicines13061408_

Round 1
Reviewer 1 Report
Comments and Suggestions for Authors
- I recommend consulting a few recent articles in the field “Araliadiol” to ensure the work is situated within the broader context of existing research. This will help in strengthening the rationale for your study.
- Comparison to current remedies, It would be attractive to match the efficacy and safety of araliadiol to existing anti-inflammatory studies. To find out the advantages and limitations of the reported work.
- Modified the conclusion based on research findings rather than general statements. Please ensure that your conclusions section emphasizes the scientific value added by your paper and the applicability of your findings. Highlight the novelty of your study. In addition to summarising the actions taken and the results, please explain their significance in the conclusions.
- It should explore the in vivo models of inflammation study for the efficacy and safety of araliadiol.
- Understand the anti-inflammatory effects of araliadiol and find potential targets for this study through mechanistic studies.
Reviewer 2 Report
Comments and Suggestions for Authors
The article entitled “Pharmacological Evaluation of Araliadiol as a Novel Anti-in-flammatory Agent in LPS-Induced RAW 264.7 Cells” presented by Seunghee Bae is a very good scientific effort. The authors have given a comprehensive evaluation of araliadiol’s anti-inflammatory effects using multiple assays, dose-dependent efficacy comparable to dexamethasone, and insights into NfκB pathway. The research topic is of high interest but some modification is required in order to enhance the quality of the paper.
- The keywords need to be reduced and also need to be in the alphabetical order.
- The introduction part needs proper modification one suggestion is to add a comparative table of known bioactive compounds from C. asiatica and their anti-inflammatory potential.
- The authors have used 1 µg/mL concentration (LPS) which is very high as usually 10–100 ng/mL concentration is mostly employed. A proper Justification is required.
- Some of the relevant references the authors need to include regarding the NF-κB pathway activation are doi: https://doi.org/10.1016/j.biopha.2020.110525, and doi: https://doi.org/10.1016/j.biopha.2017.08.120.
- Positive control needs to be incorporated in the ATP Assay.
- In figure 1, the authors need to add the error bars in Western blot.
- The calculations need to be rechecked in the figure 2 as fold-change values (e.g., IL-6: 61,465-fold) are a bit surprising.
- Loading controls are required to be added in the nuclear fractionation blots (figure 4 and figure 5).
- The authors claim of "superior efficacy to DEX" seems to be exaggerated without IC50 comparisons.
- The conclusion part needs to be rewrite and details findings of this research should be mentioned.
- The reference needs to be updated and more recent references need to be cited.
Reviewer 3 Report
Comments and Suggestions for Authors
The manuscript aims to evaluate the pharmacological properties of araliadiol, a naturally occurring polyacetylene compound, as a novel anti-inflammatory agent. The description is accurate, and the results are consistent with the conclusions. However, there are a few minor points that should be addressed:
- Provide more details about araliadiol, such as its structural formula, methods of extraction, and its potential uses.
- Discuss the limitations of this study.
- Clarify why the results of the RNA expression analysis are not confirmed by western blotting.
